# Supra-threshold vibration applied to the foot soles enhances jump height under maximum effort

Jeongin Moon[1]°, Prabhat Pathak[1]°, Sudeok Kim[1], Se-gon Roh[2], Changhyun Roh[3], Youngbo Shim[4], Jooeun Ahn[1,5]*

1 Department of Physical Education, Seoul National University, Seoul, Republic of Korea, 2 Robot Center in Samsung Seoul R&D Campus, Samsung Electronics Co., Ltd., Seoul, Republic of Korea, 3 WI Robotics, Suwon, Gyeonggi, Republic of Korea, 4 T-Robotics, Osan, Gyenggi, Republic of Korea, 5 Institute of Sport Science, Seoul National University, Seoul, Republic of Korea

° These authors contributed equally to this work.
* ahnjooeun@snu.ac.kr

**Data Availability Statement:** All relevant data are within the manuscript and its Supporting information file.

## Abstract

Previous studies have shown that absence or reduction of cutaneous sensory feedback can diminish human motor performance under maximum effort. However, it has not been explored whether any appropriate intervention in the cutaneous sensory input can augment the output motor performance, particularly in motor tasks such as jumping that involve the kinematic chain of the entire body. Using shoes with active vibrating insoles, we applied mechanical vibration to the soles of 20 young and healthy adults and evaluated the change in the jump height and muscle activation using within-participants repeated measures. The noise-like vibration having an amplitude of 130% of the sensory threshold of each participant led to an average increase of 0.38 cm in the jump height (p = 0.008) and activation of the rectus femoris of the dominant leg (p = 0.011). These results indicate that application of a properly designed cutaneous stimulus to the soles, the distal end effectors of motor tasks, can augment the output performance by involving the prime movers distant from the end effector.

## Introduction

Recent findings suggest that sensory input from the periphery not only provides information for the central nervous system but also directly affects the motor output. Multiple seminal studies have discovered distinct regions in the cerebral cortex responsible for each of sensory and motor function [1–3], but additional evidence from animal studies has suggested noticeable overlap between the two. Donoghue and Wise discovered an overlap between sensory and motor functions in the hind limb movement of rodents [4], and Matyas et al. found a direct pathway for cortical motor control driven by the primary somatosensory cortex in the mouse whisker system [5]. More recent studies have further supported the notion of possible neural flow from the primary somatosensory cortex (S1) to the primary motor cortex (M1) in rodents [6,7].

**Funding:** JA acquired all the fundings for this study, including the National Research Foundation of Korea (NRF) Grants funded by the Korean Government (MSIT) (No. 2016R1A5A1938472); the Technology Innovation Program (No. 20008912) and Industrial Technology Innovation Program (No. 20007058, Development of safe and comfortable human augmentation hybrid robot suit) funded by the Ministry of Trade, Industry & Energy (MOTIE, Korea); and the research Grant funded by the Samsung Advanced Institute of Technology, Samsung Electronics Co., Ltd. URLs. https://www.nrf.re.kr/eng/main/ https://english. motie.go.kr/www/main.do https://www.sait. samsung.co.kr/saithome/main/main.do No, The funders had no role in study design, data collection and analysis, decision to publish, or preparation of the manuscript.

**Competing interests:** JM, PP, SR, CR, YS, and JA are inventors of two patents (10-2019-0118121 and 10-2019-0170835), which were applied to Korean Intellectual Property Office by SNU R&DB Foundation. The active insole shoes used in this study were developed by the Samsung Electronics, which employs SR. These shoes might be commercialized in the future. This does not alter our adherence to PLOS ONE policies on sharing data and materials.

The existence of such an overlap and a direct pathway between the sensory and motor functions in animal models is consistent with the effect of cutaneous sensory input on the motor output of humans under maximum effort. Shim et al. clearly showed that maximum finger force decreased significantly following anesthesia applied to the fingers [8]. Seo et al. reported the effect of grip surface on maximum pinch force. They demonstrated that maximum finger force depends on the grip surface and the resulting tactile sensory input [9]. Caminita et al. observed that the effect of sensory input on human motor performance was not limited to hand-related tasks. They showed that cooling the foot soles using an ice-water bath and the resultant decrease in the cutaneous sensory feedback reduced the squat jump height [10]. Evidence from multiple motor control studies suggests a plausible mechanism for this direct effect of the sensory input on the motor output. Tactile sensory input can induce the activation of the primary somatosensory cortex, which in turn can induce the activation of the primary motor cortex [11,12], and therefore, modulation of the sensory stimulation may change the maximal force [13].

However, in contrast with the clear evidence of a decrease in the motor performance following complete removal or reduction in the sensitivity of the sensory system [10,14], there is a dearth of studies exploring methods for increasing the maximal human motor performance by augmenting the sensory feedback. In the present study, we aimed to suggest a feasible way to enhance human motor performance in a whole-body task under maximum effort using a sensory stimulus. Among various motor tasks involving whole-body dynamics, we specifically selected the squat jump to minimize motion variability due to the participant's own volition and to exploit the precision and accuracy of the available performance-measuring instrument. Employing the active insole-embedded shoes that we devised to generate mechanical vibration with a wide range of frequencies and a controlled amplitude [15], we could apply deliberately designed mechanical vibration to the soles of the participants. We hypothesized that a properly designed additional tactile sensory input from the vibrating insoles can enhance the motor output under maximum effort (the squat jump height).

## Materials and methods

### Participants

We recruited 20 healthy young adults (16 men and 4 women, age: 26.25 ± 4.01 years, height: 173.15 ± 7.58 cm, weight: 69.55 ± 12.63 kg). Based on a previous meta-analysis that addressed the effect of vibration on jump performance in young adults [16], we set the effect size as 1.00. According to the power analysis guidelines provided by Columb et al. [17], we set the p-value for statistical significance and power as 0.05 and 0.95, respectively. Subsequently, we used the G*Power software (version 3.1.9.7, Heinrich Heine University, Dusseldorf, German) and calculated a sample size of 16 according to the method suggested by Faul et al. [18]. The participants had no known history of neuromuscular, cardiovascular, or orthopedic injuries. All aspects of this study complied with the principles and guidelines described in the Declaration of Helsinki. The Institutional Review Board of Seoul National University approved the study protocol (IRB No. 1807/001-007). The participants provided written informed consent before participation. The individual whose picture is shown in this manuscript has given written informed consent (as outlined in PLOS consent form) to publish the photo of the individual.

### Equipment

Active insoles were custom-built and embedded in a pair of shoes to deliver mechanical vibrations to the soles of the participants (Fig 1). Each active insole contained two pairs of piezoelectric actuators (Disc Benders-Bimorphs, model no. 20–2225, American Piezo Ceramics Inc.,

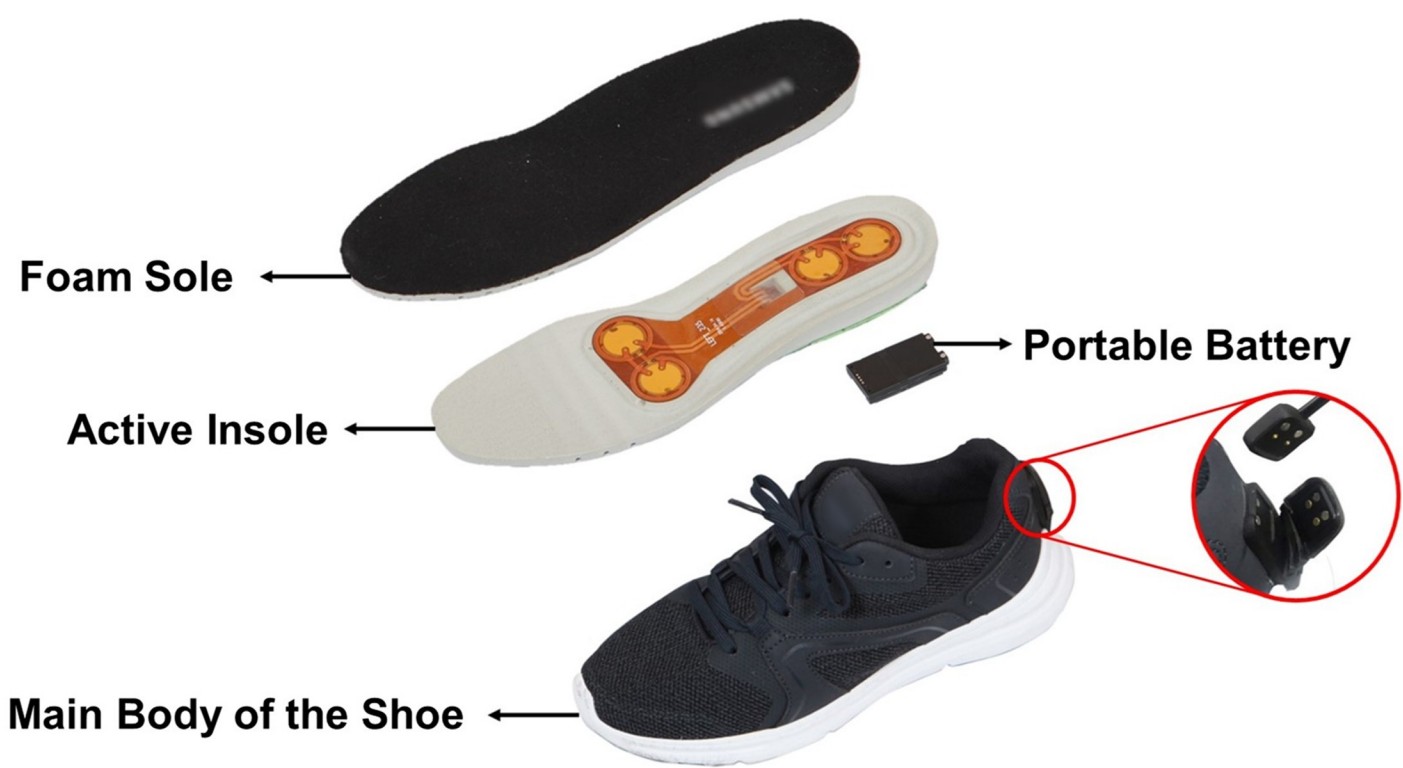

**Fig 1. The custom-built shoe with an active insole.** The system consisted of a foam sole, an active insole, a portable battery, and the main body. The portable battery could be attached to the bottom of the active insole. Mechanical vibration was applied to the metatarsal head and heel using the actuators in the active insole, which could be placed inside the shoes and connected to the charging port of the shoes. The red circle shows a magnified view of the charging port that can be opened and connected to a custom-made power cable.

Mackeyville, PA, USA) The actuators had a drive voltage of 15 V, maximum frequency of 1–350 Hz, thickness of 0.36 ± 0.10 mm, and diameter of 21 ± 0.30 mm. One pair of actuators was installed at the front of the insole and the other at the rear to provide mechanical vibration to the metatarsal head and heel, respectively. The charging port for the active insole battery was embedded in the heel cap. Using a custom-built smartphone application and Bluetooth technology, we wirelessly controlled the amplitude of the vibration of each pair of actuators to stimulate the metatarsal head or heel of each foot.

A force platform (Model FP6090-15-2000, Bertec Inc., Columbus, OH, USA) was used to record the vertical ground reaction force (VGRF) at a sampling frequency of 1000 Hz. Non-invasive surface electromyography (EMG) sensors (Trigno™ Wireless Systems, Delsys Inc., Natick, MA, USA) were used to record the muscle activations from the eight prime movers: the tibialis anterior (TA), gastrocnemius medialis (GM), rectus femoris (RF), and biceps femoris (BF) of both the legs. EMG signals were recorded at a sampling frequency of 2000 Hz and the force platform was synchronized with the EMG sensors using a sync box to match the initiation of data acquisition from both systems.

## Experimental procedure

Since the optimal vibration frequency affecting the motor performance can be different for each individual, the insole was designed to provide white noise with frequencies between 1 and 350 Hz. The amplitude of mechanical vibration was selected based on the estimated sensory threshold, the amplitude at which the participants started to sense the vibration. Both the

magnitude and the reliability of the sensory threshold estimation depend on the posture of the participant. The sensory threshold in the seated position is lower than that in the standing position [19], but the estimation of the sensory threshold during the standing position is generally less reliable due to the unintended shifting of weight [20]. Therefore, we selected the seated position for higher reliability and asked the participants to sit upright with equal weight on each foot. The sensory thresholds at the metatarsal head and heel were estimated separately, since the sensitivity depends on the region of the foot sole [19]. We incrementally increased the amplitude of vibration and asked the participants to report as soon as they felt the stimulation. We repeated this process of sensory threshold estimation three times each for the metatarsal head and heel of both the feet in a random order for each participant. The sensory threshold for each area of the foot was calculated as the mean value of the three estimates. We selected the amplitude of vibration as 130% of the sensory threshold of each foot area of each participant based on the results of a previous study that reported 130% of the sensory threshold as the optimal amplitude of supra-threshold vibration that can enhance the motor performance during a mildly difficult postural control task [21].

Each participant performed seven trials of the vertical squat jump under two experimental conditions: vibration at 130% of the sensory threshold (ON) and no vibration (OFF). The sequences of the 14 trials (7 ON and 7 OFF) were randomized for each participant. The overall procedure for the jump task is depicted in Fig 2. We instructed the participants to stand on the force platform with their hands to the side and to place their feet shoulder-width apart. Once the experimenter provided a ready signal, the participants bent their knees and lowered their body so that their fingers touched the front section of the shoes. The participants were instructed to remain as still as possible in this position. After approximately 3 seconds, the experimenter gave a verbal signal to jump by saying "jump" and the participants jumped as high as possible, extending the arms upward.

## Data processing

Vertical jump height (VJH) was calculated by integrating the VGRF obtained from the force platform [22]. Initially, the raw VGRF signal was filtered using a zero-lag second-order low-pass Butterworth filter with a cut-off frequency of 10 Hz. Subsequently, we inspected the VGRF time profile to calculate the body weight (BW) and the impulse generated by the VGRF during jump motion. The overall process for selecting the reference points for the necessary calculations is illustrated in Fig 3. The moment of vertical jump takeoff was defined as the moment when the foot left the ground or when the VGRF value reached its minimum. We tagged this moment $t_1$ as a datum point. Subsequently, we identified a section that started at 3 s before $t_1$ and ended at $t_1$. We divided this section of 3 s into time bins of 50 ms. We searched for a time bin with the least standard deviation (SD) of the VGRF and defined the BW of a participant as the mean VGRF during that specific time bin. We also defined the initial moment of the jump motion ($t_0$) as the moment after which the VGRF began to be equal to or lower than 99% of the BW for the second last time within the 3-s interval ending at $t_1$. If the VGRF profile had no local minimum below 99% of the BW, $t_0$ was defined as the moment after which the VGRF began to be equal to or larger than 101% of the BW for the last time within the 3-s interval.

According to the linear momentum principle, the impulse generated during the jump motion becomes

$$L = \int_{t_0}^{t_1} (VGRF - BW)dt = mv_1 \tag{1}$$

where $v_1$ is the vertical speed of a participant's center of mass (COM) at $t_1$ and $m$ is the mass of

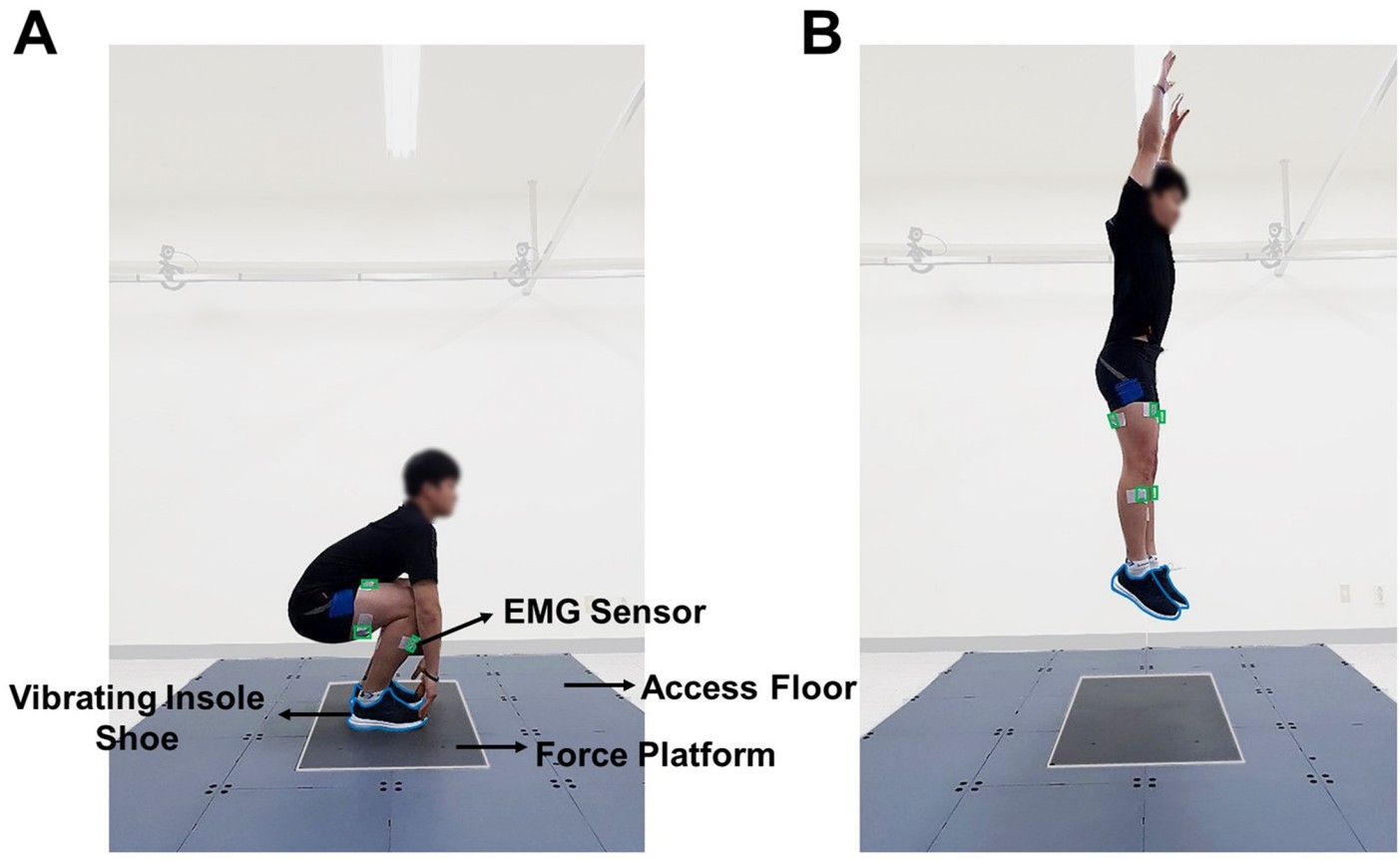

**Fig 2. Demonstration of the vertical jump task.** (A) The initial position of the participant on the force platform before the jump. Placing their feet shoulder-width apart, participants bent their knees and lowered their bodies so that their fingers touched the toe section of the shoes. (B) The highest position after the experimenter gave the participants a verbal signal to jump. The participants jumped as high as possible, extending their arms upward. Eight non-invasive EMG sensors were used to measure the muscle activation levels of the tibialis anterior, gastrocnemius medialis, rectus femoris, and biceps femoris of both the legs. The placement sites of the EMG sensors, the vibrating shoe, and the force platform are highlighted in green, blue, and light gray colors, respectively.

the subject (BW divided by $g$, the gravitational acceleration). Disregarding the air drag, gravity is the only force acting on the body of a participant during the flight phase. Hence, the COM of the participant satisfies

$$\frac{1}{2}mv_1^2 + mgy_1 = \frac{1}{2}mv_2^2 + mgy_2 \qquad (2)$$

where $y_1$ is the height of the COM at $t_1$ and $v_2$ and $y_2$ are the vertical speed and height of the COM, respectively, at an arbitrary time point $t_2$. The jump height $y_2$ reaches the maximum value when all the kinetic energy is converted to potential energy or when $v_2$ becomes zero. In other words,

$$VJH = \max(y_2 - y_1) = v_1^2/2g \qquad (3)$$

Substituting Eq (1) into Eq (3),

$$VJH = L^2/2m^2g \qquad (4)$$

The EMG signals from the TA, GM, RF, and BF of both the limbs were recorded. We particularly investigated the activation of these muscles during the time interval of the jump

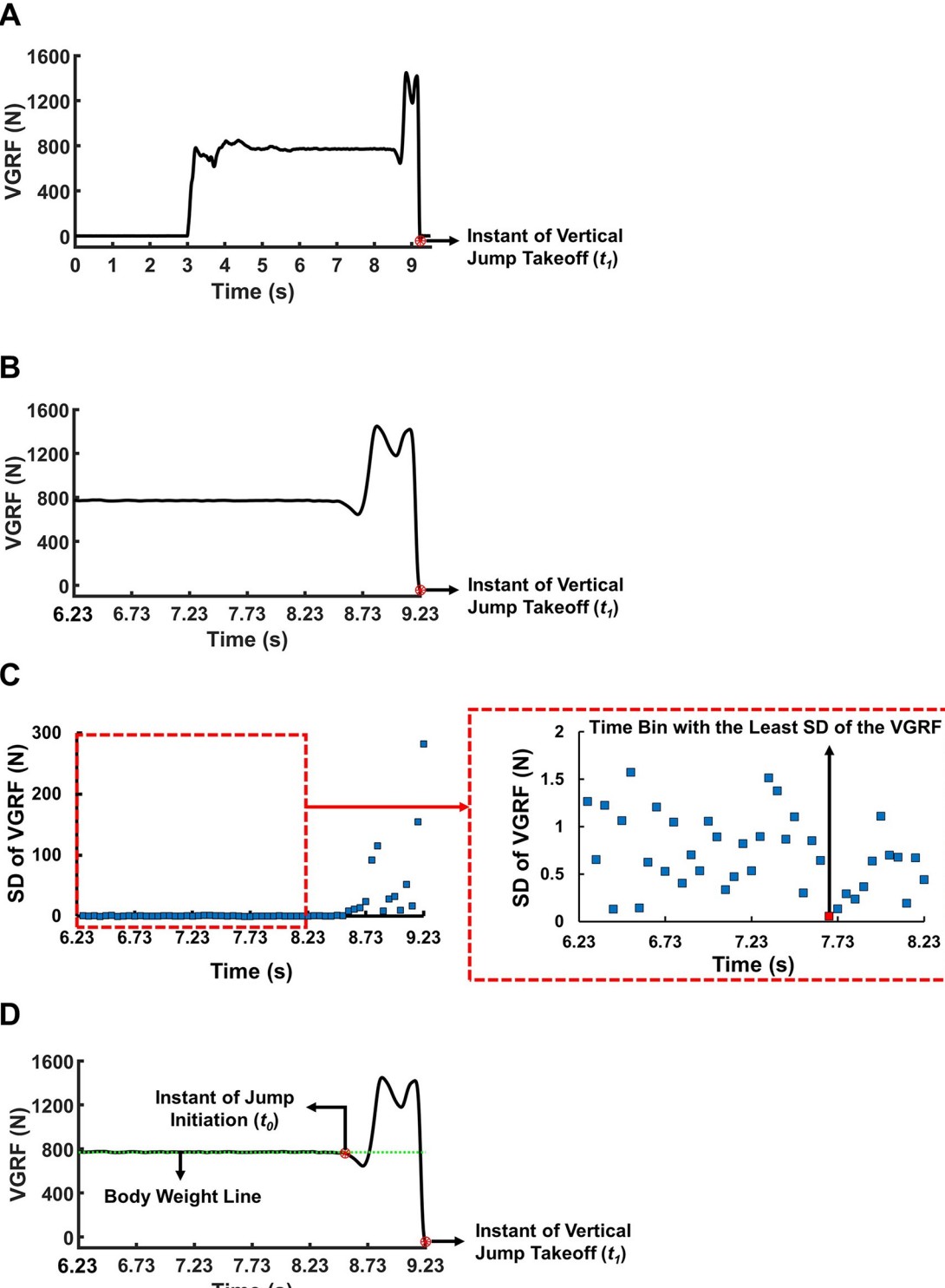

**Fig 3. An illustration of the overall process of selecting the reference points for the calculation of vertical jump height (VJH) of a representative participant.** (A) The time profile of the vertical ground reaction force (VGRF) during a single trial of the jumping task. The moment of vertical jump takeoff was defined as the moment when the VGRF value reached its minimum ($t_1$) (denoted by a red star inside a red circle). (B) The section of the VGRF time profile that started at 3 s before $t_1$ and ended at $t_1$. This section was used to calculate the body weight (BW) of the participants. (C) The standard deviation (SD) of the VGRF. The VGRF section in (B) was divided into time bins of 50 ms (denoted by blue squares). The SD of the VGRF during each time bin was

calculated. The initial 2 s of the section are enlarged on the right side to illustrate the time bins more clearly. The time bin with the least SD of the VGRF is highlighted by a red square. The BW was estimated as the mean VGRF during the red time bin. (D) The moment of jump initiation ($t_0$). The moment after which the VGRF began to be equal to or lower than 99% of the BW for the second last time within the VGRF time profile in (B) was selected as $t_0$ (denoted by a red star in a red circle). The green line denotes the BW of the participant. The integration of the difference between the VGRF and BW from $t_0$ to $t_1$ was defined as the net impulse during the jump motion, which was used to calculate the VJH.

motion (from $t_0$ to $t_1$). To remove any noise from the electrical devices, a notch filter at 60 Hz was applied to the full-wave rectified raw EMG signals. A zero-lag fourth-order band-pass filter with a cut-off frequency between 20 Hz and 350 Hz was then used to remove movement artifacts and high-frequency noise from the EMG signals. A moving average filter having a time window of 1% of the length of the EMG signals for each trial was additionally applied for further smoothing [23]. Finally, the smoothed EMG signals were integrated over time to calculate the index of the overall activation of each muscle (integrated EMG, or IEMG).

$$IEMG = \int_{t_0}^{t_1} EMG \, dt \qquad (5)$$

## Statistical analysis

A paired t-test was used to evaluate any significant differences in the VJH, time interval of the jump motion, average VGRF during the jump, and IEMG of the prime movers of the left and right legs between the ON and OFF conditions. For a sanity check, the BW and the VGRF variability were also compared statistically between the two conditions. The Shapiro–Wilk test was performed to test the normality of the data. Statistical significance was set at $p < 0.05$.

## Results

The means and standard errors (SEs) of the VJHs of 20 participants under the two experimental conditions (ON and OFF) are shown in Fig 4A. The results of the paired t-test revealed that the mean VJHs of the participants during the ON condition were significantly higher than those during the OFF condition; t (19) = 2.987, p = 0.008. This increase in the jump height or the impulse resulted from increased VGRF rather than from an increase in the duration of ground contact (Fig 4B and 4C). The paired t-test did not show significant differences in the duration of the contact time before takeoff between the two experimental conditions; t (19) = -1.600, p = 0.126, whereas the average VGRFs during the ON condition were significantly higher than those during the OFF condition; t (19) = 2.870, p = 0.010. We also compared the IEMG values of the prime movers of the dominant and non-dominant legs between the two experimental conditions. The paired t-test concluded that the IEMGs of the RF of the dominant leg during the ON condition were significantly higher than those during the OFF condition; t (19) = 2.806, p = 0.011 (Fig 4D). The average IEMGs of the TA of both the legs and RF of the non-dominant leg were also higher under the ON condition, although the difference was not statistically significant.

Since the intervention adopted in this study was mechanical vibration and the output performance was estimated based on the resultant VGRF, it might be plausible to argue that the applied vibration itself affected the measurement. To address this issue, we investigated whether the estimated BW, which was used to calculate the VJH, remained approximately constant between the ON and OFF conditions. The means and SEs of the BWs under the two

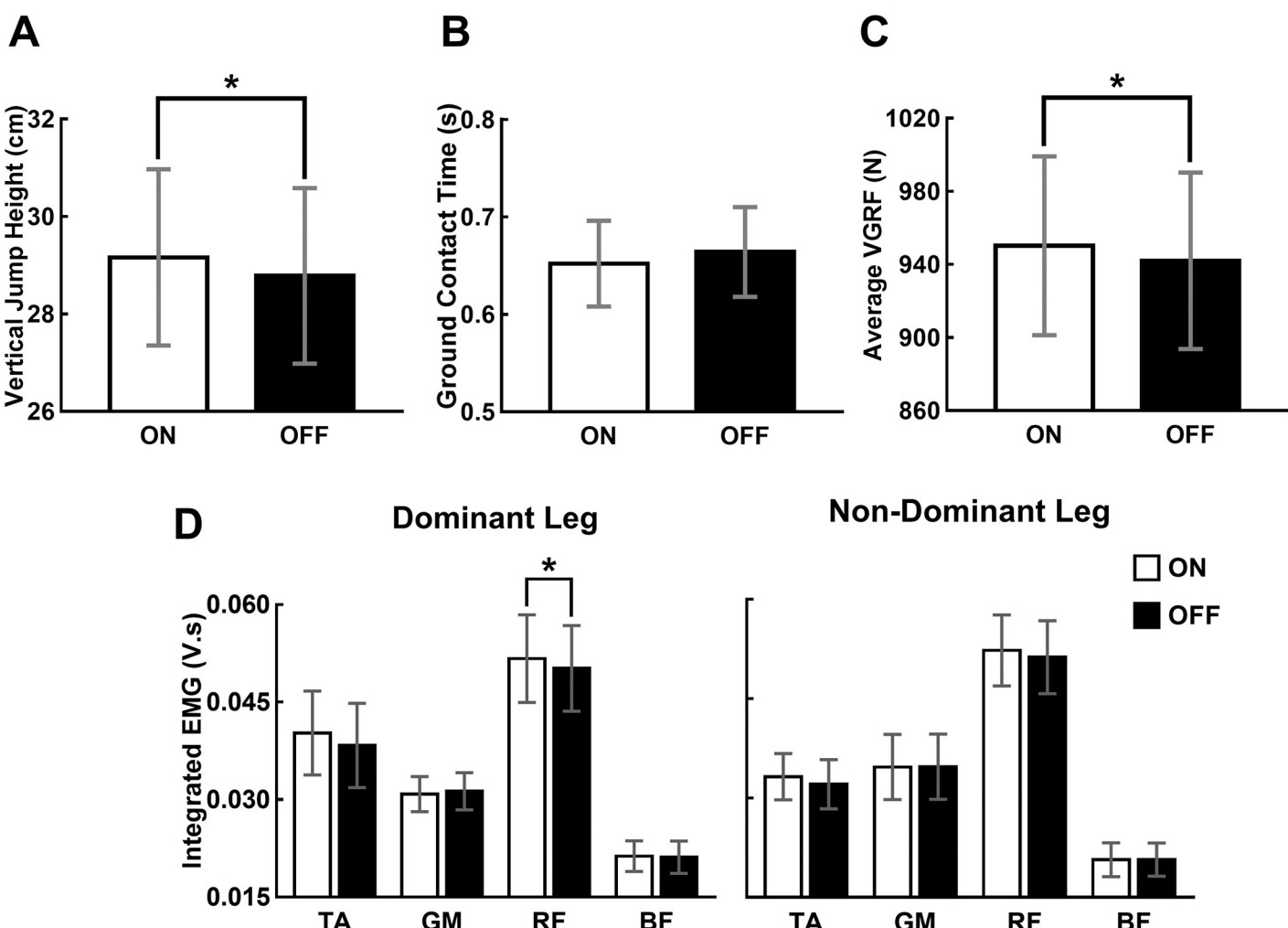

**Fig 4. The effect of active insole vibration.** (A) The effect of vibration on the vertical jump heights (VJHs) of 20 participants under two experimental conditions (ON: Vibration at 130% of the sensory threshold, OFF: No vibration). The paired t-test concluded that vibration increased the VJH. (B) The effect of vibration on the ground contact duration before takeoff. (C) The effect of vibration on the average vertical ground reaction force (VGRF) during the ground contact before takeoff. (D) The effect of vibration on the integrated electromyography (EMG) of the prime movers (tibialis anterior [TA], gastrocnemius medialis [GM], rectus femoris [RF], and biceps femoris [BF]). The error bars and * denote standard errors and statistically significant difference (p < 0.05), respectively.

experimental conditions are shown in Fig 5A. The paired t-test revealed no statistically significant difference in the BW between the two conditions; t (19) = 0.101, p = 0.920.

We also calculated the SD of the VGRF during the 50-ms time bin used to calculate the BW for each participant and each trial. The SD distribution of the VGRF during the time bin with the least variability was compared statistically between the two experimental conditions (Fig 5B). The paired t-test revealed no statistically significant difference in the SD of the VGRF during the time bin between the two conditions; t (19) = 0.710, p = 0.487. These results indicate that mechanical vibration did not significantly affect the VGRF variability or the reliability of the measured BW and VGRF.

## Discussion

The present study showed that a properly designed cutaneous stimulus can increase the maximal squat jump height. Multiple studies have supported the direct connection between sensory

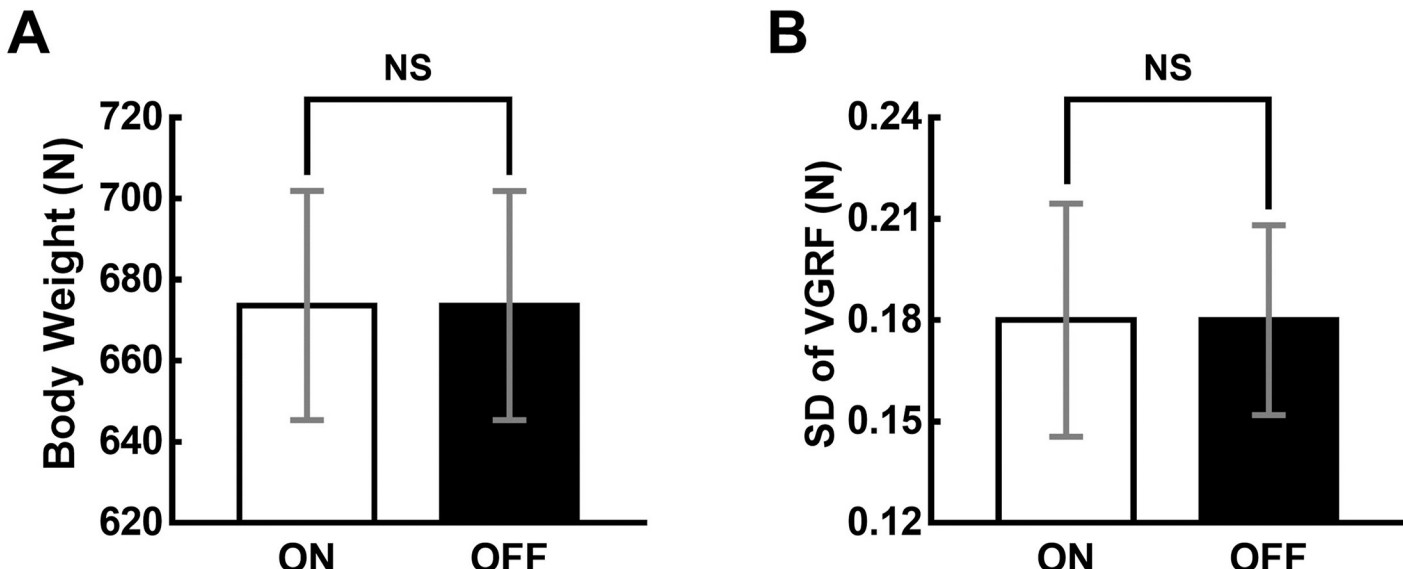

**Fig 5. The body weight (BW) and standard deviation (SD) of the vertical ground reaction force (VGRF) during the BW calculation phase.** (A) The BWs of 20 participants under the two experimental conditions (ON: Vibration at 130% of the sensory threshold, OFF: No vibration). (B) The SD of the VGRF during the BW calculation phase of 20 participants under the two experimental conditions. The error bars and NS denote standard errors and no statistically significant difference (p > 0.05), respectively.

and motor systems by demonstrating the effect of reduction in the cutaneous input on motor performance under maximum effort. However, most of the studies have focused on finger or hand-related movement [8,9,24]; whether a change in the amount of sensory input can also affect the maximal motor output of the whole-body movement remains to be clarified. Caminita et al. showed that cooling-induced decrease in the sensitivity of the soles reduced the squat jump height [10], but it remains unclear how much of the reduction in the jump height results from the reduction in the sensory input rather than reduction in the dexterity of the plantar fascia and other toe-related muscles whose control is critical in the jumping motion [25,26]. More importantly, although previous studies have focused on decreased maximal motor output due to reduction in the sensory input, an increase in the motor performance under maximum effort due to any sensory input augmentation has been seldom investigated. Using the active insole-embedded shoes that we developed, we generated mechanical vibration that added sensory input and demonstrated that this stimulus increased the jump height of young and healthy adults.

We quantified the jump height by integrating the VGRF with respect to time and calculating the initial takeoff velocity rather than by measuring the kinematics. This approach enabled us to measure the jump height with high resolution by exploiting the precision and the fast-sampling rate of the force platform. Therefore, an increase in the estimated jump height directly indicates an increase in the impulse, and our results showed that this increased impulse was attributed to an increase in the muscle force (Fig 4A–4C). We did not measure the force from each individual muscle, but the observed increase in the average VGRF, which is the reaction force of the net force exerted by the participants on the ground, indicates an increase in the muscle force. Although a significant change in the activation level was observed only in the RF of the dominant leg (Fig 4D), increased RF activation partly explains the increased jump height considering its important role in the squat jump [27].

The original rationale behind hypothesis formulation for this study was the previously observed direct effect of the sensory input on the motor output; we expected that the "sensory-

to-motor overflow [10]" would allow the additional cutaneous input of vibration to augment the motor output. However, vibration-induced increases in muscle activation and jump height have also been reported in previous studies in the field of sports science and medicine. Recently, mechanical vibration has been adopted to improve athletic performance during training. Vibration can be applied to a specific muscle group directly using small actuators or to the whole body of a person standing on a vibrating platform. Both methods induce the tonic vibration reflex (TVR), which increases the muscle force through excitation of the primary muscle spindles [28,29]. Multiple studies have reported an increase in the muscle strength, peak force, and rate of force development following direct muscle vibration [30,31], and application of whole-body vibration has also been reported to improve the VJH, muscle strength, and power capabilities [32–34].

However, the intervention adopted in this study was clearly different from the vibration applied to local muscles or the stimulus from a vibrating platform. The active insole did not directly vibrate the muscles that contributed to the vertical jump. Although the insoles applied mechanical vibration to the soles of the participants as a vibrating platform does, the amplitude of the insole vibration we applied in this study was only 130% of the sensory threshold. Note that the typical amplitude of a vibrating platform is 2–4 mm [34–36], whereas the typical vibration detection threshold in humans is between 18 nm and 25 μm [37]; the amplitude of the stimulus of a vibrating platform is larger than that applied in the present study by an order of magnitude of 100 or more. Thus, the ratio between the mechanical power transmitted by the active insole and the vibrating platform is approximately 1/10,000 or lower, since the power transmitted by a wave is proportional to the square of its amplitude. These factors suggest that it is highly unlikely that the increase in the jump height due to insole vibration has the same mechanism as that due to a vibrating platform; the mechanical power transmitted by the vibrating insole would not be sufficient to induce the TVR.

On the other hand, the results of the present study are consistent with the direct effect of the sensory input on the motor output. Tactile sensory input is associated with activation of the primary somatosensory cortex, which can send inputs to the primary motor cortex [11,12]. Hence, modulation of sensory stimulation can lead to a change in force production [13]. It is also noteworthy that cutaneous interventions were not directly applied to the prime muscles for finger movements in previous studies that investigated the effect of tactile input on finger forces [8,9]. The tactile interventions were applied only to the most distal "end effector" of motor function such as the skin of the fingertips. The results of these previous studies are consistent with the results of the present study. The amplitude of vibration of the active insole was set at 130% of the sensory threshold; it might not be high enough to induce TVR, but it was high enough to increase the sensory input. Application of such vibration most likely stimulated mechanoreceptors such as Pacinian and Meissner corpuscles, which are sensitive to low-amplitude vibration [38]. Hence, the increased sensory input induced by stimulation of these mechanoreceptors could enhance the activation of the primary motor cortex and increase the output force and jump height. Furthermore, as in the cases of the previous studies that addressed the effect of sensory input on the finger force [8,9], the vibration stimulus did not directly excite the prime muscles involved in the jump motion in the present study. We applied the tactile sensory input only to the sole, which is the end effector responsible for direct mechanical interaction between the human body and the ground. This sensory stimulus to the end effector resulted in a noticeable change in the motor output of the whole body (the jump height).

In this initial study, we used active insoles to apply the designed vibration to the soles without elaborate tuning of the stimulus. Previous studies have reported that the effect of vibration on the motor output depends on the frequency and amplitude of the stimulus [30,39].

Moreover, the damping effect of the skin depends on the thickness of the skin or fat. Thus, it is highly plausible that the optimal parameters of vibration differ across individuals. The cumbersome procedure of selecting such optimal parameter values was deliberately omitted in the present study to avoid excessive experiment duration. Instead, we systematically estimated the sensory threshold of each participant and selected 130% of the threshold as the amplitude of vibration. To determine the level of vibration stimulus, we consulted the only available previous study that investigated the effect of various amplitudes of vibration on motor performance; Severini and Delahunt demonstrated that stimulation with 130% of the threshold improved balancing performance most effectively [21]. However, jumping and balancing are distinct tasks, and therefore, it is unknown that 130% is also the most effective level for enhancing jump performance. In terms of frequency of vibration, we opted to apply white noise that contains a wide range of frequencies instead of finding and applying the optimal frequency, which may also differ across individuals. Even with this simple and crude procedure for parameter selection, the designed vibration enhanced the jump height. A future work of more detailed tuning of the vibration parameters might further enhance the motor performance.

## Supporting information

**S1 File. Manuscript data.** The obtained data of the vertical jump heights, vertical ground reaction forces (VGRFs), body weights, standard deviation of VGRFs, duration of jump motions and integrated electromyography.
(XLSX)

## Author Contributions

**Conceptualization:** Jooeun Ahn.

**Data curation:** Prabhat Pathak.

**Formal analysis:** Prabhat Pathak.

**Funding acquisition:** Jooeun Ahn.

**Investigation:** Jeongin Moon, Prabhat Pathak, Sudeok Kim.

**Methodology:** Jeongin Moon, Prabhat Pathak, Jooeun Ahn.

**Project administration:** Jooeun Ahn.

**Resources:** Jeongin Moon, Prabhat Pathak, Se-gon Roh, Changhyun Roh, Youngbo Shim.

**Software:** Jeongin Moon, Prabhat Pathak, Se-gon Roh, Changhyun Roh, Youngbo Shim.

**Supervision:** Jooeun Ahn.

**Validation:** Jooeun Ahn.

**Visualization:** Jeongin Moon, Prabhat Pathak.

**Writing – original draft:** Jeongin Moon, Prabhat Pathak, Jooeun Ahn.

**Writing – review & editing:** Jooeun Ahn.

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
