## [Decision Letter · Decision Letter 0]

8 Feb 2022

PONE-D-21-29963Supra-threshold vibration applied to the foot soles enhances jump height under maximum effortPLOS ONE

Dear Dr. Ahn,

Thank you for submitting your manuscript to PLOS ONE. After careful consideration, we feel that it has merit but does not fully meet PLOS ONE’s publication criteria as it currently stands. Therefore, we invite you to submit a revised version of the manuscript that addresses the points raised during the review process.

We look forward to receiving your revised manuscript.

Kind regards,

Shazlin Shaharudin

Academic Editor

PLOS ONE

Journal Requirements:

JM, PP, SR, CR, YS, and JA are inventors of two patents (10-2019-0118121 and 10-2019-0170835), which were applied to Korean Intellectual Property Office by SNU R&DB Foundation. The active insole shoes used in this study were developed by the Samsung Electronics, which employs S.R., C.R., and Y.S. These shoes might be commercialized in the future.

3. We note that Figure 1 in your submission contain copyrighted images. All PLOS content is published under the Creative Commons Attribution License (CC BY 4.0), which means that the manuscript, images, and Supporting Information files will be freely available online, and any third party is permitted to access, download, copy, distribute, and use these materials in any way, even commercially, with proper attribution. For more information, see our copyright guidelines: http://journals.plos.org/plosone/s/licenses-and-copyright.

4. We note that Figure 2 includes an image of a  participant in the study]. 

Reviewers' comments:

Reviewer's Responses to Questions

**Comments to the Author**

1. Is the manuscript technically sound, and do the data support the conclusions?

Reviewer #1: Yes

Reviewer #2: Yes

2. Has the statistical analysis been performed appropriately and rigorously? 

Reviewer #1: No

Reviewer #2: Yes

3. Have the authors made all data underlying the findings in their manuscript fully available?

Reviewer #1: Yes

Reviewer #2: Yes

4. Is the manuscript presented in an intelligible fashion and written in standard English?

Reviewer #1: Yes

Reviewer #2: Yes

5. Review Comments to the Author

Reviewer #1: Dear author,

Line 62: citation needed.

Line 111-116: citation needed, & need to explain the procedure of markers placement

Line 216-219: citation needed on the procedure filtering and smoothing.

Line 223 - Results: How many t-test was conducted? It seems many paired t-test thus increase chances of Type 1 error. Need to rethink better options of analysis (e.g. 1 WAY RM ANOVA).

Reviewer #2: This paper presents the proof-of-concept of custom-built active vibrating insoles in shoes with mechanical vibration delivered to evaluate the change in the jump height and muscle activation. Vertical jump height, body weight and impulse generated during jump motion were calculated.

For suggestion, it is worthwhile to mention why the amplitude of insole vibration applied was 130% of the sensory threshold. Eventhough this value is based on a previous study, are there any similarities or differences between this study and the previous research? Does this value based on the design of the product or necessary for the jump activity?

6. PLOS authors have the option to publish the peer review history of their article (what does this mean?). If published, this will include your full peer review and any attached files.

Reviewer #1: No

Reviewer #2: No

---

## [Author Response · Author response to Decision Letter 0]

11 Mar 2022

Editor

Thank you for your advice. We revised the manuscript and file names according to PLOS ONE's style requirements.

JM, PP, SR, CR, YS, and JA are inventors of two patents (10-2019-0118121 and 10-2019-0170835), which were applied to Korean Intellectual Property Office by SNU R&DB Foundation. The active insole shoes used in this study were developed by the Samsung Electronics, which employs S.R., C.R., and Y.S. These shoes might be commercialized in the future.

Two of the authors (CR and YS) changed their affiliations, and we updated the manuscript and Competing Interests section accordingly. We confirm that the situation clarified in the Competing Interests statement does not alter our adherence to PLOS ONE policies on sharing data and materials. Following the editor’s suggestion, we also included our updated Competing Interests statement in the cover letter.

3. We note that Figure 1 in your submission contain copyrighted images. All PLOS content is published under the Creative Commons Attribution License (CC BY 4.0), which means that the manuscript, images, and Supporting Information files will be freely available online, and any third party is permitted to access, download, copy, distribute, and use these materials in any way, even commercially, with proper attribution. For more information, see our copyright guidelines: http://journals.plos.org/plosone/s/licenses-and-copyright.

We (the authors) took Figure 1 by ourselves; we are the original copyright holder. Considering the Editor’s concern and following the suggestion from a PLOS ONE journal staff, Rose Ann Joyce Sagun Puetes, we additionally made the commercial logo on the tongue of the shoe invisible. We now confirm that no copyrighted material is present in Figure 1.

4. We note that Figure 2 includes an image of a participant in the study]. 

Thank you for your advice. We received written informed consent (PLOS consent form) from the participant shown in Figure 2. We also added the statement about the participant’s consent for publication in the revised manuscript (lines 90-91).

In the revision, we added a few references to address comments from the reviewers. We rechecked all the updated references and found that no reference has been retracted.

Reviewer #1

Dear author,

Line 62: citation needed.

Thank you for your suggestion. In the revision, we cited the reference showing that removed or reduced sensitivity decreases the motor performance (line 64).

Line 111-116: citation needed, & need to explain the procedure of markers placement.

That paragraph explains the equipment and methods we used in our own study. To clarify further, no marker set was applied because we did not record kinematics. Instead, we calculated vertical jump height (VJH) of each subject based on the impulse generated during jump motion and the subject’s body weight (BW).

Line 216-219: citation needed on the procedure filtering and smoothing.

In the revision, we added the reference for the procedure of EMG signal filtering and smoothing (line 221).

Line 223 - Results: How many t-test was conducted? It seems many paired t-test thus increase chances of Type 1 error. Need to rethink better options of analysis (e.g. 1 WAY RM ANOVA).

We understand that Type-I error could be inflated in determining the significance among three or more conditions and it should be corrected by the Bonferroni adjustment for pairwise comparison (MacDonald, P.L., & Gardner, R.C., Edu. Psychol. Meas. 60, 735-754 (2019)). However, in this study, we aimed to compare outcome variables under “two conditions”: with vibration and without vibration. To recap our study, we performed one paired t-test to check the effect of vibration on the vertical jump height, and then, we performed another paired t-test to check the effect of vibration on the muscle activation level. Therefore, in the case of our study, the two independently performed t-tests do not affect the chance of Type-I error.

Reviewer #2 

This paper presents the proof-of-concept of custom-built active vibrating insoles in shoes with mechanical vibration delivered to evaluate the change in the jump height and muscle activation.

Vertical jump height, body weight and impulse generated during jump motion were calculated.

For suggestion, it is worthwhile to mention why the amplitude of insole vibration applied was 130% of the sensory threshold. Eventhough this value is based on a previous study, are there any similarities or differences between this study and the previous research? Does this value based on the design of the product or necessary for the jump activity?

We appreciate reviewer’s comment and suggestion. As we explained in the manuscript, we chose the amplitude of the vibration as 130% of the threshold considering a previous study that demonstrated the efficacy of such vibration during a mildly difficult balancing task. However, this choice was not based on any similarity between jumping and balancing. (Although both jumping and balancing require whole body activation, it is definitely arguable that they are similar.) It was neither based on the design of the product; the used insole can apply the vibration from 0 to about 200% of the sensory threshold of most participants with fine resolution.

So, let us explain the rationale behind 130% more clearly in this response. First, the main motivation of the study was the recently observed “sensory-to-motor overflow.” Thus, we aimed to apply detectable sensory stimulation rather than sub-threshold vibration; we needed to apply vibration with the amplitude of 110, 120, 130, 140%, etc. of the threshold. Second, we aimed to investigate whether the additional detectable tactile stimulation can increase the motor performance under maximal effort. Among various motor tasks that require maximal effort, we chose jump, considering the convenience and reliability of the measure of the motor output. However, unfortunately, we could not find any similar previous study that investigated the effect of vibration amplitude on “jump” performance; this fact also makes this study as a unique contribution to this field. The only study that investigated the effect of various amplitudes of detectable vibration on motor task was what we cited, so we simply consulted the “best available” reference and chose 130%, which had been reported as the most effective level of the stimulus. So, the value of 130% was not based on either similarity between jumping and balancing (if any) or the design of the used insole. It was determined based on (limited) available data and reference. Considering reviewer’s comment, we revised the manuscript to clarify this (lines 371-376). 

Despite this rationale, it is never guaranteed that our choice of 130% is the most effective level; rather, it is highly plausible that optimal parameters differ across individuals. In this initial study, we only demonstrated that some detectable vibration, a certain level of one kind of additional sensory input, can increase the motor performance under maximal effort. Tuning of the parameter of such stimulation requires much more future work. Considering reviewer’s comment, we revised the manuscript to further clarify this limitation and the necessary future work (lines 379-380).

---

## [Editor Report · Decision Letter 1]

24 Mar 2022

Supra-threshold vibration applied to the foot soles enhances jump height under maximum effort

PONE-D-21-29963R1

Dear Dr. Ahn,

We’re pleased to inform you that your manuscript has been judged scientifically suitable for publication and will be formally accepted for publication once it meets all outstanding technical requirements.

Kind regards,

Shazlin Shaharudin

Academic Editor

PLOS ONE
---

## [Editor Report · Acceptance letter]

29 Mar 2022

PONE-D-21-29963R1 

Supra-threshold vibration applied to the foot soles enhances jump height under maximum effort 

Dear Dr. Ahn:

I'm pleased to inform you that your manuscript has been deemed suitable for publication in PLOS ONE. Congratulations! Your manuscript is now with our production department. 

Kind regards, 

on behalf of

Dr. Shazlin Shaharudin 

Academic Editor

PLOS ONE